# CONCENTRATED ATTENTION FOR MULTI-AGENT REINFORCEMENT LEARNING

## ABSTRACT

In cooperative multi-agent reinforcement learning, centralized training with decentralized execution (CTDE) shows great promise for a trade-off between independent Q-learning and joint action learning. However, vanilla CTDE methods assumed a fixed number of agents could hardly adapt to real-world scenarios where dynamic team compositions typically suffer from the dilemma of dramatic partial observability variance. Specifically, agents with extensive sight ranges are prone to be affected by trivial environmental substrates, dubbed the "attention distraction" issue; ones with limited observability can hardly sense their teammates, hindering the quality of cooperation. In this paper, we propose a Concentrated Attention for Multi-Agent reinforcement learning (CAMA) approach, which roots in a divide-and-conquer strategy to facilitate stable and sustainable teamwork. Concretely, CAMA targets dividing the input entities with controlled observability masks by an Entity Dividing Module (EDM) according to their contributions for attention weights. To tackle the attention distraction issue, the highly contributed entities are fed to an Attention Enhancement Module (AEM) for execution-related representation extraction via action prediction with an inverse model. For better out-of-sight-range cooperation, the lowly contributed ones are compressed to brief messages by a Attention Replenishment Module (ARM) with a conditional mutual information estimator. Our CAMA outperforms the SOTA methods significantly on the challenging StarCraftII, MPE, and Traffic Junction benchmarks.

## 1 INTRODUCTION

Cooperative multi-agent deep reinforcement learning (MARL) has gained increasing attention in many areas such as games (Berner et al., 2019; Samvelyan et al., 2019; Kurach et al., 2019), social science (Jaques et al., 2019), sensor networks (Zhang & Lesser, 2013), and autonomous vehicle control (Xu et al., 2018). With practical agent cooperation and scalable deployment capability, centralized training with decentralized execution (CTDE) (Rashid et al., 2018; Gupta et al., 2017) has been widely adopted for

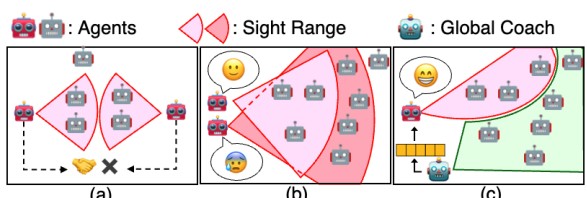

Figure 1: The dynamic sight range dilemma. (a) Agents can hardly cooperate beyond their sight ranges. (b) Agents with large sight ranges may perform worse due to "attention distraction". (c) A sketch of our CAMA.

MARL. Current CTDE methods usually assume a fixed number of agents such as QMIX (Rashid et al., 2018), MADDPG (Lowe et al., 2017), QPLEX (Wang et al., 2020a), etc. To adapt to complicated and dynamic real-world scenarios with dynamic team compositions (i.e., the team size varies), researchers extend these methods by introducing the attention mechanism (Vaswani et al., 2017), which usually requires splitting the state of the environment into a series of entities (Yang et al., 2020; Agarwal et al., 2019; Iqbal et al., 2021).

However, attention-based methods can hardly handle the varying partial observability (e.g., the varying sight range of each agent) in multi-agent systems, Fig. 1. With severe partial observability, agents usually lose the sight of teammates, leading to the poor coordination quality. We use a demo in Sec. 5.1 to verify the phenomenon. With slight partial observability (large sight ranges with

near perfect information), these methods exhibit apparent performance degradation that more trivial entities may distract the agents' attention and interfere with their decision making. See Sec. 4.1 for a detailed analysis. Therefore, maintaining agents' attention on potential cooperators and execution-related entities to adapt to the partial observability variation is crucial for MARL in challenging environments.

In this paper, we propose a Concentrated Attention for Multi-Agent reinforcement learning (CAMA) approach, which roots in a divide-and-conquer strategy to facilitate stable and sustainable teamwork via attention learning. Specifically, we first use an Entity Dividing Module (EDM) to divide the raw entities into two parts for each agent according to its attention weights. For the attention distraction issue in settings with large sight ranges, an Attention Enhancement Module (AEM) is applied on the entities with high attention weights for execution-related representation extraction via action prediction with an inverse model. For out-of-sight-range coordination in low sight ranges, an Attention Replenishment Module (ARM) with a novel conditional mutual information estimator is applied to compress the information in entities with low attention weights. With the above three modules, agents' attention can be properly concentrated on execution of local actions and potential teamwork to deal with dynamic partial observability.

We evaluate our method on three commonly used benchmarks: StarCraftII (SC2) (Samvelyan et al., 2019), Multi-agent Particle Environment (MPE) (Lowe et al., 2017), and Traffic Junction (Sukhbaatar et al., 2016). The proposed CAMA outperforms SOTA methods significantly on all conducted experiments and exhibits remarkable robustness to sight range variation and dynamic team composition.

## 2 RELATED WORK

As a popular paradigm for single reward MARL, CTDE is a trade-off between independent Q-learning (Tan, 1993) and joint action learning (Claus & Boutilier, 1998). Centralized training makes agents cooperate better while decentralized execution benefits the flexible deployment capability. A series of works concentrate on distributing the team reward to all agents by value function factorization (Sunehag et al., 2018; Rashid et al., 2018), deriving and extending the Individual-Global-Max (IGM) principle for policy optimality analysis (Son et al., 2019; Wang et al., 2020a; Rashid et al., 2020; Wan et al., 2021). To avoid constraints of IGM, some works then delve into applying centralized critics on local policies using the actor-critic paradigm (Lowe et al., 2017; Foerster et al., 2018; Zhou et al., 2020). Although CTDEs have achieved great progresses in recent years, with fixed sizes of agents, they are typically impeded by the dynamic team composition issue (Schroeder de Witt et al., 2019; Liu et al., 2021) in real-world applications.

**Dynamic Team Composition**. When the agent number varies in each episode, the attention mechanism (Vaswani et al., 2017) is commonly adopted to handle the issue (Jiang et al., 2018; Agarwal et al., 2019; Yang et al., 2020; Hu et al., 2020; Iqbal et al., 2021). Some works develop a set of curricula to adapt to the increasing team sizes (Baker et al., 2019; Long et al., 2020; Wang et al., 2020c) with non-negligible computational costs for training on different team sizes. Iqbal et al. (2021) add auxiliary Q-learning tasks to increase the multi-agent system's robustness by randomly masking out part of agents' observability, which increases the types of situations encountered by agents. Although these methods adapt to different team sizes well, they still suffer obvious attention distraction when the sight ranges of agents is large, and are prone to fail in some situations where agents with limited observability must cooperate beyond their sight ranges (Liu et al., 2021).

**Communication Mechanism** is a feasible solution to enhance agents' cooperation. Recently some works regard the relationship between agents as a proximity-based or fully-connected graph and assume the information can propagate among the graph edges (Foerster et al., 2016; Suttle et al., 2020; Agarwal et al., 2019; Zhang et al., 2018; Sukhbaatar et al., 2016; Liu et al., 2020; Mao et al., 2020a). These methods usually let agents communicate with all neighbors or the whole team, which brings high communication costs. Moreover, the communication mode is sensitive to team sizes. Some works assume the existence of a centralized coach to integrate information and send messages to all agents (Liu et al., 2021; Mao et al., 2020b; Niu et al., 2021), which requires centralized execution, with relatively low communication costs. The communication messages are usually trained by backpropagation of RL loss, sometimes with constraint from mutual information objective to reduce the communication bandwidth (Wang et al., 2020b), help action decision (Yuan et al., 2022)

or predict future trajectories (Liu et al., 2021). Unlike these methods, we use communication from a centralized coach to apply attention replenishment for agents' better coordination.

## 3 BACKGROUND

**MARL Symbols.** We model a fully collaborative multi-agent task with $n$ agents as a *decentralised partially observable Markov decision process* (Dec-POMDP) (Oliehoek et al., 2016) $G = \langle S, A, I, P, r, Z, O, n, \gamma \rangle$, where $s \in S$ is the environment's state. At time step $t$, each agent $i \in I_a \equiv \{1, ..., n_a\}$ chooses an action $a^i \in A$, which makes up the joint action $\mathbf{a} \in \mathbf{A} \equiv A^{n_a}$. $P(s_{t+1}|s_t, \mathbf{a}_t) : S \times \mathbf{A} \times S \to [0, 1]$ is the environment's state transition distribution. All agents share the same reward function $r(s, \mathbf{a}) : S \times \mathbf{A} \to \mathbb{R}$. The discount factor is denoted by $\gamma \in [0, 1)$. Each agent $i$ has its local observations $o^i \in O$ drawn from the observation function $Z(s, i) : S \times I \to O$ and chooses an action by its stochastic policy $\pi^i(a^i|\rho^i, \chi^i) : \Gamma \times X \to \Delta([0, 1]^{|A|})$, where $\rho^i \in \Gamma \equiv (O \times A)^l$ denotes the action-observation history of agent $i$, and $l$ is the number of state-action pairs in $\rho^i$. $\boldsymbol{\rho}$ is the action-observation histories of all agents. $\chi^i \in X$ denotes the additional communication message and $\pi^i$ has no dependence on $\chi^i$ in CTDE. The agents' joint policy $\pi$ induces a joint *action-value function*: $Q^\pi(s_t, \mathbf{a}_t) = \mathbb{E}_{s_{t+1:\infty}, \mathbf{a}_{t+1:\infty}}[R_t|s_t, \mathbf{a}_t]$, where $R_t = \sum_{k=0}^{\infty} \gamma^k r_{t+k}$ is the discounted accumulated team reward. The goal of MARL is to find the optimal joint policy $\pi^*$ such that $Q^{\pi^*}(s, \mathbf{a}) \geq Q^\pi(s, \mathbf{a})$, for all $\pi$ and $(s, \mathbf{a}) \in S \times \mathbf{A}$.

**Value Function Factorization.** During execution in CTDE, each agent chooses actions by its local $Q$ function $Q^i(\rho^i)$ induced by local observation $o^i$. During training, since all agents share a common team reward, a global $Q$ function $Q^{tot}$ is calculated from all local $Q^i$ conditioned on the global state $s$ by a $Mixer$ module as: $Q^{tot}(\boldsymbol{\rho}, \mathbf{a}) = g(Q^1(\rho^1, a^1), ..., Q^n(\rho^n, a^n), s)$. To guarantee the global optimality from the local optimality, a Mixer should satisfy the IGM principle (Son et al., 2019): $\arg\max_{\mathbf{a}} Q^{tot}(\boldsymbol{\rho}, \mathbf{a}) = (\arg\max_{a^1} Q^1(\rho^1, a^1), ..., \arg\max_{a^n} Q^n(\rho^n, a^n))$.

**Multi-Head Attention (MHA) in MARL.** Under the condition of dynamic teams, the observation vector for each agent may have varying sizes during one episode, and therefore we can hardly apply traditional methods which only accept fixed-size input. In contrast, we use the "entity-wise input" to represent the observation and the multi-head attention (MHA) module to embed the entities with dynamic number into a fixed length vector for each agent. To feed MHA, the raw state $s$ of the environment is commonly expressed as a series of entities $e^i$, i.e., $s^e := \{e^i\}, i \in [1, n_e]$ with the same vector length, where $n_e$ is the maximum number of entities. The entities include agents we can or can not control, and other substrates in the multi-agent scenario (e.g. obstacles). We consider a proximity-based observation function $Z(s^e, i)$ for agent $i$, i.e., each agent has a sight range $SR$, and $Z(s^e, i) := \{e^j | d(i, j) <= SR\}, j \in [1, n_e]$, where $d(i, j)$ is the Euclidean (or Manhattan) distance between entity $i$ and $j$. Let $\mathcal{X} \in \mathbb{R}^{n_e \times d}$ be the entity input, in which each row is an entity. Let $I_a \subseteq I := \{1, ..., n_e\}$ be the set of indices that selects which entities of the input $\mathcal{X}$ are used to compute queries such that $\mathcal{X}_{I_a} \in \mathbb{R}^{n_a \times d}$ (usually $i \in I_a$ means entity $i$ is an controllable agent, and $I_a := \{1, ..., n_a\}$). The attention head is as follows:

$$\text{AH}\left(I, \mathcal{X}, \mathcal{M}; \boldsymbol{W}^Q, \boldsymbol{W}^K, \boldsymbol{W}^V\right) = \text{softmax}\left(\text{mask}\left(\frac{\boldsymbol{Q}\boldsymbol{K}^\top}{\sqrt{h}}, \mathcal{M}\right)\right)\boldsymbol{V} \in \mathbb{R}^{|I| \times h},$$
$$\boldsymbol{Q} = \mathcal{X}_{I_a}\boldsymbol{W}^Q, \boldsymbol{K} = \mathcal{X}\boldsymbol{W}^K, \boldsymbol{V} = \mathcal{X}\boldsymbol{W}^V, \quad \mathcal{M} \in \{0, 1\}^{n_a \times n_e}, \boldsymbol{W}^Q, \boldsymbol{W}^K, \boldsymbol{W}^V \in \mathbb{R}^{d \times h}. \tag{1}$$

The $\text{mask}(\mathcal{Y}, \mathcal{M})$ operation takes two matrices with the same size as input, and fills the entries of $\mathcal{Y}$ with $-\infty$ where $\mathcal{M}$ equals 0. If we set the values in the positions of unseen entities to 0, this operation blocks the information from certain entities after softmax, to uphold the partial observability for local agents. $\boldsymbol{W}^Q, \boldsymbol{W}^K$, and $\boldsymbol{W}^V$ are learnable parameters. Then we can define the mulit-head attention module by concatenating $n^h$ attention heads together:

$$\text{MHA}(I, \mathcal{X}, \mathcal{M}) = \text{concat}\left(\text{AH}\left(I, \mathcal{X}, \mathcal{M}; \boldsymbol{W}_j^Q, \boldsymbol{W}_j^K, \boldsymbol{W}_j^V\right), j \in \left(1 \ldots n^h\right)\right), \tag{2}$$

where information not blocked can be integrated across all entities.

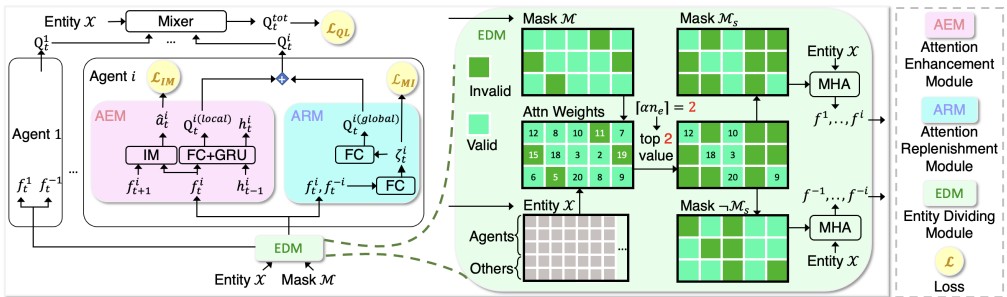

Figure 2: Network structure of our proposed CAMA. Entities and the observation mask are first fed into the Entity Dividing Module (EDM) to get the Attention Enhancement embedding $f^i$ and Attention Replenishment embedding $f^{-i}$, which is trained by inverse model (IM) loss and mutual information (MI) objective, respectively. For RL training, a local $Q^i$ is generated by $f^i$, $f^{-i}$, and its observation-action history with a GRU, and further fed into the mixing network for $Q^{tot}$.

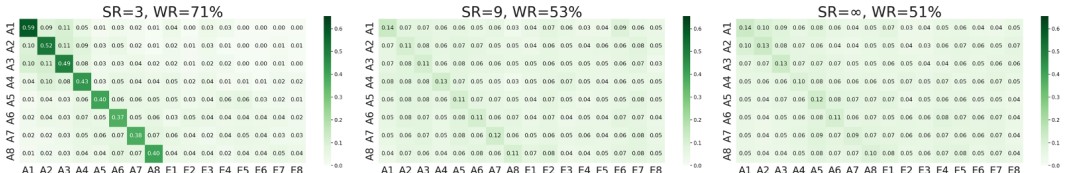

Figure 3: The attention distraction issue illustration with the agents' average attention heatmap of REFIL on StartCraftII map "3-8sz_symmetric" (the hardest scenario "8sz_vs_8sz") during one episode on all entities. The y axis denotes 8 agents (A1-A8), while the x axis with additional 8 enemies (E1-E8) records 16 entities. The sum of each row is normalized to 1. "SR" is each agent's sight range, and "WR" is the winning rate against preset AI.

## 4 METHOD

### 4.1 INTUITION AND OVERVIEW

**Intuition.** The performance of traditional MARL methods is highly affected by the partial observability of the environment. We use the agents in a game (e.g., StarCraftII) for example. When the sight range is small, the agents can hardly find the teammates and support them, leading to poor team coordination (a demo in Sec. 5.1 verifies this hypothesis). However, it is counterintuitive that with increasing sight ranges, the agents' performance typically degrades (this phenomenon is detailed in Sec. 5.1). We argue that the agents' attention is easily distracted by unrelated scenarios, causing the *attention distraction* issue. To reveal the importance of agents' attention on their performance, we train the SOTA algorithm REFIL (Iqbal et al., 2021) on different sight ranges (3,9, and $\infty$) and visualize the agents' attention weights on all entities (i.e., the value of the matrix $\boldsymbol{QK}^\top$ in MHA module) in Fig. 3. It can be seen that when more entities are visible, the agents' attentions get more dispersed so that more difficulties need to be overcome to win the preset AI. Existing dynamic team MARL methods commonly face such problem (see Appendix E).

To deal with the dilemma of the dynamic partial observability in MARL, we resort to a divide-and-conquer learning strategy, dubbed CAMA, for stable performance and sustainable teamwork. Concretely, in low sight ranges we need to improve agents' attention on realizing the potential cooperators by out-of-sight-range information, while in large sight ranges, agents' attention concentration on execution-related entities should be kept.

**Framework and RL Training.** CAMA mainly consists of three components, including an Entity-Dividing Module (EDM) for dynamic partial observability control, an Attention Enhancement Module (AEM) for attention concentration on execution-related entities, and an Attention Replenishment Module (ARM) for agents' out of sight range coordination, Fig. 2. Specifically, for each agent $i$, EDM divides and embeds the raw entities as $f^i$ and $f^{-i}$ to feed AEM and ARM respectively. For AEM, an inverse model is applied to resolve the attention distraction issue. For ARM, a coach with global

sights is introduced to generate a communication message $\zeta^i$ for agents' coordination. Following Yuan et al. (2022), we generate a $Q^{i(local)}$ from $f^i$ and agent $i$'s observation-action history $\rho^i$ (which is the output of a gated recurrent unit (GRU) cell), and a $Q^{i(global)}$ from $\zeta^i$. $Q^i$ is computed by their summation, i.e., $Q^i = Q^{i(local)} + Q^{i(global)}$. All local $Q^i$s are fed into a mixing network (Yang et al., 2020) to calculate a $Q^{tot}$. The RL loss can be formulated as follows:

$$\mathcal{L}_{QL} = \mathbb{E}_{(\mathbf{a}_t, r_t, \boldsymbol{\rho}_t, \boldsymbol{\rho}_{t+1}) \sim \mathcal{D}} \left[ \left( r_t + \gamma \max_{\mathbf{a}'} \hat{Q}^{tot}(\boldsymbol{\rho}_{t+1}, \mathbf{a}') - Q^{tot}(\boldsymbol{\rho}_t, \mathbf{a}_t) \right)^2 \right], \tag{3}$$

where $\hat{Q}^{tot}$ is the target network, and $\mathcal{D}$ is the replay buffer.

## 4.2 Entity Dividing Module

An EDM divides raw entities into an attention enhancement part and an attention replenishment part by their ranking of attention weights. For the former one, we wish to constrain the maximum of observed entities to avoid the attention distraction. And the latter one should contain enough out-of-sight-range information for team coordination. We first deal with the attention enhancement part. Recall that in MHA module, $\mathcal{M} \in \{0,1\}^{n_a \times n_e}$ is a binary mask applied on the entity embeddings which is generated by the environment. To uphold each agent's partial observability, a more sparse mask $\mathcal{M}_s$ is introduced to replace $\mathcal{M}$ which satisfies the following constraints:

$$||\mathcal{M}_s||_\infty \leq \alpha n_e, \neg \mathcal{M}_s \odot \neg \mathcal{M} = \neg \mathcal{M}, \tag{4}$$

where $n_e$ is the maximum number of entities, $\alpha \in (0,1]$ is a hyper-parameter, and $\odot$ means the element-wise multiplication operation. The negation of $\mathcal{M}$ is defined as $\neg \mathcal{M} := \mathbf{1} - \mathcal{M}$, where $\mathbf{1}$ is an all 1 matrix with the same shape as $\mathcal{M}$. The left side of Eq. (4) ensures that the percent of observable entities is less than $\alpha$, while the right side makes the agent observe the entities available in the original mask $\mathcal{M}$ only. We can assign a low value for $\alpha$ (e.g., 0.4) to limit each agent's visible entities in complicated environments. To get $\mathcal{M}_s$, we define a $\mathcal{B}_i(\boldsymbol{W})$ operation to get the *indices* of the top $i$ values in each row of $\boldsymbol{W}$. $\mathcal{M}_s$ can be calculated as follows:

$$\mathcal{M}_s = \mathcal{M} \odot \mathcal{M}_f, \mathcal{M}_f[\boldsymbol{I}] = 1, \mathcal{M}_f[\text{others}] = 0, \boldsymbol{I} = \mathcal{B}_{\lfloor \alpha n_e \rfloor}(\boldsymbol{Q}\boldsymbol{K}^\top), \tag{5}$$

where $\lfloor \cdot \rfloor$ means to round down the value, and $\boldsymbol{Q}$, $\boldsymbol{K}$ are matrices of queries and keys in MHA module, respectively. Under Eq. (5), each agent only remains the sight on at most $\lfloor \alpha n_e \rfloor$ entities with the highest attention weight, and improves its attention concentration. We prove in Appendix A that $\mathcal{M}_s$ obtained by Eq. (5) is adequate for Eq. (4). After getting $\mathcal{M}_s$, we can compute $f^i$ for the attention enhancement part, which is the output of $\text{MHA}(I, \mathcal{X}, \mathcal{M}_s)$.

Since the attention replenishment part should contain all the information not involved in the former one for better out-of-sight-range coordination, we transmit it with $f^{-i}$, which is the embedding of the complement entities of $\mathcal{M}_s$ on $s$ with the same MHA module as $\text{MHA}(I, \mathcal{X}, \neg \mathcal{M}_s)$.

## 4.3 Attention Enhancement for Local Agent

When knowing what will happen when a specific action is taken, the learned agents can hardly be distracted. Thereby, we aim to concentrate agents' attention on execution-related information distilled from the high-dimensional state space. Specifically, we resort to the inverse model (Pathak et al., 2017), a two-layer MLP, that uses the local observation $o_t^i$ and $o_{t+1}^i$ to predict the agent $i$'s action $a^i$. In the prediction module, the local observation $o_t^i$ and $o_{t+1}^i$ are first fed into the same EDM to get the features $f_t^i$ and $f_{t+1}^i$. Then, the probability of each action can be predicted as $p(\hat{a}_t^i) = IM(f_t^i, f_{t+1}^i; \theta)$. The learning loss can be defined as:

$$\mathcal{L}_{IM} = CE(p(\hat{a}_t^i), a_t^i), \tag{6}$$

where $CE$ means the cross entropy. $f^i$ will contain the necessary information for predicting $a^i$ by optimizing Eq. (6), which encourages EDMs to discard irrelevant distracting information for the consciousness enhancement embedding $f^i$. With the auxiliary representation learning task, the learned embedding $f^i$ can be used to calculate each agent's local $Q$ function $Q^{i(local)}$.

## 4.4 ATTENTION REPLENISHMENT BY GLOBAL COACH

To equip agents with the ability of out-of-sight-range coordination, we use a centralized coach equipped with global states to generate a message $\zeta^i$ from $f^{-i}$ and compute a $Q^{i(global)}$ for each agent $i$ at each time step (Liu et al., 2021; Niu et al., 2021; Mao et al., 2020b). The message plays the role of attention replenishment when agents facing difficulty to cooperate through their local observations. The learning objective for $\zeta^i$ should contain the information unknown to agent $i$, and not distract the agent's attention. Therefore, we maximize:

$$\mathcal{I}(\zeta^i; f^{-i}) - \beta\mathcal{I}(\zeta^i; s) = (1 - \beta)\mathcal{I}(\zeta^i; s) - \mathcal{I}(\zeta^i; f^i | f^{-i}), \tag{7}$$

where $s$ is the global state, and $\mathcal{I}(\cdot; \cdot)$ means the mutual information. $f^{-i}$ indicates the replenishment information for agent $i$. We leave the derivation of Eq. (7) to Appendix B. Maximizing $\mathcal{I}(\zeta^i; f^{-i})$ let $\zeta^i$ be the summary of $f^{-i}$. By feeding $\zeta^i$ to agent $i$, the agent can sense the information beyond its sight range, and further alleviate the difficulties of cooperation caused by partial observability. Minimizing $\mathcal{I}(\zeta^i; s)$ compresses the information $\zeta^i$ has, which can be regarded as an information bottleneck constraint on $\zeta^i$ (Wang et al., 2020b). We use a hyper-parameter $\beta \in [0, 1]$ to control the compression degree of $\zeta^i$. Combining the two terms, we (1) discard the information in $f^i$, which is already known to agent $i$, and (2) compress the sophisticated $f^{-i}$ into a brief message, which can hardly distract the agent while promotes coordination.

We then separately optimize the two mutual information term in the right side of Eq. (7). Directly maximizing $\mathcal{I}(\zeta^i; s)$ is difficult, but there exist some tools estimating its differentiable lower bound, e.g. infoNCE (Oord et al., 2018) and MINE (Belghazi et al., 2018). We choose the CatGen formulation (Fischer, 2020) of the former one and maximize the following lower bound of $\mathcal{I}(\zeta^i; s)$:

$$\bar{\mathcal{I}}_{NCE}(\zeta^i; s) = -\mathbb{E}_{\zeta^i, s}[\log \frac{p(\zeta^i|s)}{\frac{1}{K}\sum_{k=1}^{K} p(\zeta^i|s^k)}], \tag{8}$$

where $K$ is the sample number of a mini-batch.

Then we move on to minimizing $\mathcal{I}(\zeta^i; f^i | f^{-i})$. There are existing tools that minimize the upper bound of the mutual information between two random variables, such as CLUB (Cheng et al., 2020) and L1Out (Poole et al., 2019). But these methods can not be directly applied to the conditional mutual information paradigm. Therefore, we extend the CLUB estimator to the conditional form and present Conditional-CLUB (CC) estimator as a differentiable upper bound of $\mathcal{I}(\zeta^i; f^i | f^{-i})$:

$$\mathcal{I}_{CC}(\zeta^i, f^i | f^{-i}) = \mathbb{E}_{\zeta^i, f^i, f^{-i}}[\log p(\zeta^i|f^i, f^{-i})] - \mathbb{E}_{f^i}\mathbb{E}_{\zeta^i, f^{-i}}[\log p(\zeta^i|f^i, f^{-i})]. \tag{9}$$

**Theorem 4.1.** *For three random variables $\zeta^i$, $f^i$ and $f^{-i}$,*

$$\mathcal{I}_{CC}(\zeta^i, f^i | f^{-i}) \geq \mathcal{I}(\zeta^i, f^i | f^{-i}). \tag{10}$$

*The equality holds if and only if $f^i$ is independent of the joint distribution of $\zeta^i, f^{-i}$.*

The proof of Thm. 4.1 can be referred in Appendix C. Accordingly, we can minimize the conditional mutual information via minimizing $\mathcal{I}_{CC}$. In practice, assuming we have the conditional distribution $p(\zeta^i|f^i, f^{-i})$, we can sample pairs $\{(\zeta_k^i, f_k^i, f_k^{-i})\}_{k=1}^{K}$ and get an unbiased estimation of $\mathcal{I}_{CC}$:

$$\hat{\mathcal{I}}_{CC} = \frac{1}{K}\sum_{k=1}^{K} \log p(\zeta_k^i|f_k^i, f_k^{-i}) - \frac{1}{K^2}\sum_{k=1}^{K}\sum_{j=1}^{K} \log p(\zeta_k^i|f_j^i, f_k^{-i}). \tag{11}$$

Since the second part of Eq. (11) requires $\mathcal{O}(K^2)$ computational complexity, we use the following faster counterpart:

$$\bar{\mathcal{I}}_{CC} = \frac{1}{K}\sum_{k=1}^{K} \log p(\zeta_k^i|f_k^i, f_k^{-i}) - \frac{1}{K}\sum_{k=1}^{K} \log p(\zeta_k^i|f_{\mu(k)}^i, f_k^{-i}), \tag{12}$$

where $\mu(\cdot)$ is a mapping from $k \in \{1, .., K\}$ to its random permutation. Since $\hat{\mathcal{I}}_{CC}$ and $\bar{\mathcal{I}}_{CC}$ are both unbiased estimators, $\mathbb{E}[\hat{\mathcal{I}}_{CC}] = \mathbb{E}[\bar{\mathcal{I}}_{CC}] = \mathcal{I}_{CC}$. In practice, we assume $p(\zeta^i|f^i, f^{-i})$ as a Gaussian distribution. We use a neural network with input $\text{concat}(f^i, f^{-i})$ to calculate its mean and variance and optimize it with the reparameterization trick (Kingma & Welling, 2013). Since $\text{concat}(f^i, f^{-i})$

has involved all the information in the global state $s$, we can use the same distribution to represent $p(\zeta^i|s)$ in $\bar{\mathcal{I}}_{NCE}$ and therefore optimizing $\mathcal{L}_{MI} = \bar{\mathcal{I}}_{CC} - (1 - \beta)\bar{\mathcal{I}}_{NCE}$. If we need a coach with stronger capability by e.g., introducing the memory module, we will get the prior distribution $p(\zeta^i|\boldsymbol{\rho})$ instead of $p(\zeta^i|f^i, f^{-i})$, therefore we can not estimate $\bar{\mathcal{I}}_{CC}$ directly. In Appendix D we introduce a variational distribution to estimate $p(\zeta^i|f^i, f^{-i})$ when it is unknown.

# 5 EXPERIMENTS

## 5.1 DILEMMA IN SIGHT RANGES

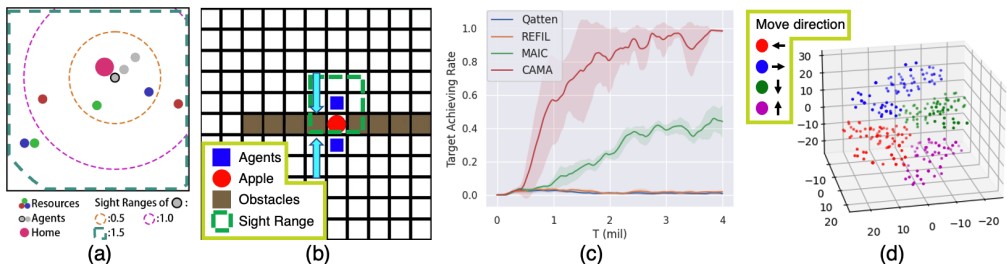

Figure 4: (a) A sketch of environment "Resource Collection".(b) A sketch of the demo "Catch Apple". (c) Task sovling rate in "Catch Apple". Qatten and REFIL are CTDE, and MAIC and CAMA have a centralized coach. (d) Visualization of messages received (grouped by the next action agents take).

**Coordination in Low Sight Ranges.** We first show the defect of CTDE methods with a demo. In a $11 \times 11$ grid-world, an apple is uniformly generated at one grid of the $3 \times 3$ grids in the map center, with obstacles of length $4$ at its right&left or up&down, Fig. 4(b). 2 controllable agents are initialized at random places. They can move towards $4$ directions, one grid at a time. The goal is that two agents should arrive the apple simultaneously, rewarded 10. Any agent touching the apple or team reaching the episode time limit will end the game. The reward of one agent touching apple is $1$. The reward for time penalty is $-0.1$. The agents can only see the entities in the $3 \times 3$ grids, each centered on its position. Agents can not overlap, which means the only way to achieve the team goal is the two agents moving towards the apple from the opposite sides of the obstacles, at which time they can not see each other. We train two CTDE methods: Qatten (Yang et al., 2020) and REFIL (Iqbal et al., 2021), and two communication-based methods: our CAMA and MAIC (Yuan et al., 2022) (with a global coach, see Sec. 5.2 for details). The target achieving rate during training are shown in Fig. 4(c). Only methods with communication can solve the task while CTDE methods fail completely.

To validate the effect of communication, we use t-SNE (Van der Maaten & Hinton, 2008) to visualize the messages $\zeta^i$ from 160 testing episodes of our method in 3D-space. We show the results of the last time step of each episode in Fig. 4(d). Each point is a message and the color means the agent's action after it receives the message. The results show that agents' actions have obvious correlation with messages received, which shows the importance of communication in out-of-sight coordination.

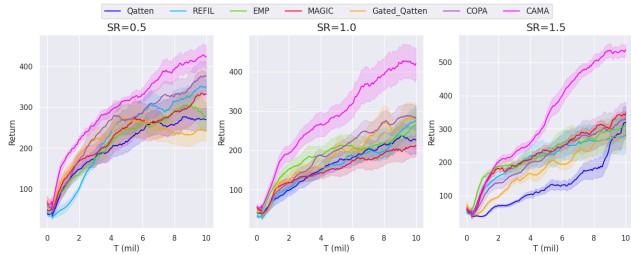

Figure 5: Test returns of 3 sight ranges (SR) on RC.

| $\mathcal{L}_{IM}$ | $\mathcal{L}_{MI}$ | $SR = 0.5$ | $SR = 1$ |
|---|---|---|---|
| | | 260.47±37.45 | 339.88±21.44 |
| | ✓ | 343.73±52.26 | 393.14±23.01 |
| ✓ | | 280.36±27.96 | 393.93±32.1 |
| ✓ | ✓ | 385.25±25.61 | 458.13±46.93 |

Table 1: Test returns on 2 sight ranges (SR) on RC. "✓" means the corresponding loss is added.

**Attention Distraction in Large Sight Ranges.** We use the Resource Collection (RC) task (Fig. 4(a)) to show the attention concentration effect of CAMA. In this task, a group of 3-8 agents coordinate to collection resources from various places and bring them home. It is slightly modified from Liu et al. (2021) to make it suitable for both CTDE and communication-based settings. Please refer to

Appendix G.2 for a detailed description. We test the performance of the following methods on three sight ranges in Fig. 5: (1) 2 CTDE methods: Qatten (Yang et al., 2020), REFIL (Iqbal et al., 2021). (2) 1 graph communication method: EMP (Agarwal et al., 2019). (3) 4 centralized communication methods: MAGIC (Niu et al., 2021), Gated_Qatten (Mao et al., 2020b), COPA (Liu et al., 2021), and our CAMA. MAGIC and Gated_Qatten are modified to make them suitable for the entity-wise input setting. The results show that CAMA outperforms all baselines in all sight range settings. And only CAMA performs better in $SR = 1.5$ than $SR = 1.0$ and $SR = 0.5$, while others suffer from attention distraction and perform worse in larger sight range.

## 5.2 COMPONENT ANALYSIS

**Entity Dividing Module.** In EDM, the parameter $\alpha$ plays the role of balancing the observability between AEM and ARM. We explore how to choose $\alpha$ in different sight ranges in Fig. 6. To make the error bars (std) more clear, points with the same $\alpha$ are slightly offset on the X axis. Note that $\alpha = 1.0$ means no constraint on observability function, i.e., deleting EDM. We find that as the sight range increases, the agents can see more entities, and therefore a lower $\alpha$ can help attention concentration.

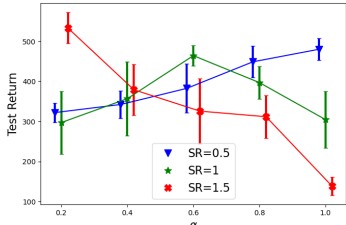

Figure 6: The Effect of $\alpha$ on different sight ranges (SR).

| $\beta$ | 1l-TJ | 2l-TJ | RC-SR=1 | RC-SR=0.5 |
|---|---|---|---|---|
| 0.1 | **30.89±2.54** | 95.54±5.73 | 389.47±41.88 | 366±15.35 |
| 0.3 | 29.9±2.99 | **99.64±4.45** | 398.74±68.33 | 379.8±36.11 |
| 0.5 | 29.89±3.21 | 95.85±3.49 | **458.13±46.93** | 385.25±25.61 |
| 0.7 | 27.58±1.23 | 91.32±0.74 | 373.19±25.43 | **396±63.19** |
| 0.9 | 27.29±0.74 | 91.28±4.23 | 354.06±20.24 | 369.4±52.52 |

Table 2: The effect of $\beta$ on different tasks: 1-lane and 2-lane Traffic Junction, Resource Collection with sight range (SR) 1 and 0.5. From left to right, the task gets harder.

**Attention Enhancement.** We analyze the contribution of each loss in Table 1. $\mathcal{L}_{IM}$ (AEM) is important when SR is large, which is reasonable since large SR brings attention distraction issue and requires agents to focus attention on execution-related entities.

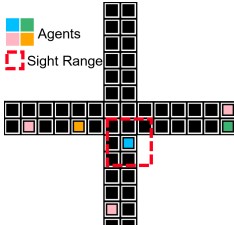

Figure 7: Traffic Junction teaser (SR=1).

| Method | MAIC&NDQ | IMAC | COPA | CAMA |
|---|---|---|---|---|
| Map   SR | $\uparrow \mathcal{I}(\zeta^i; a^i \mid s)$ | $\downarrow \mathcal{I}(\zeta^i; o^i)$ | $\uparrow \mathcal{I}(\zeta^i_t; \rho^i_{t+T}, s_t)$ | $\uparrow \mathcal{I}(\zeta^i; f^{-i}) \downarrow \mathcal{I}(\zeta^i; s)$ |
| 1-lane   0 | 23.35±4.45 | 19.67±0.43 | 17.38±2.92 | **30.18±2.94** |
| 1-lane   1 | 21.81±3.04 | 20.75±0.37 | 20.39±0.5 | **29.89±3.21** |
| 2-lane   0 | 66.81±12.33 | 65.74±6.59 | 73.16±12.37 | **96.28±7.64** |
| 2-lane   1 | 64.62±8.81 | 58.5±2.27 | 59.23±4.68 | **95.85±3.49** |

Table 3: Results on Traffic Junction. $SR = 0$ or $1$ means the agents can see themselves only or can observe the $3 \times 3$ grids around them. "$\uparrow$" means MI maximization and "$\downarrow$" means minimization.

**Attention Replenishment.** Table 1 shows that $\mathcal{L}_{MI}$ (ARM) can bring obvious improvement under both SR conditions. To check whether the improvement comes from the communication mechanism or mutual information objective, we turn to a grid-world based traffic junction environment (TJ). We use the map from Sukhbaatar et al. (2016), which is a crossroads where cars are continuously generated from a Poisson distribution at one of the four entrances, and aiming for one of the other three exits (Fig. 7). Unlike the original simplified setting that the routes are fixed so that agents only choose to accelerate or brake, we use a more difficult setting that agents can move towards four directions freely, which is harder and more realistic. Please refer to Appendix G.2 for environment details. We compare four MI-based message generators under our entity-wise setting: MAIC&NDQ (Yuan et al., 2022; Wang et al., 2019), IMAC (Wang et al., 2020b), COPA (Liu et al., 2021) and our CAMA. To make a fair comparison on MI objective, we only implement the mutual information part of each method and keep the others same to ours, e.g., the structure of neural networks and communication mechanism. Since MAIC and NDQ use similar MI objectives under the centralized coach setting, we regard them as one method. The test return results are shown in Table 3, which exhibit the superior performance of our MI objective in limited sight range.

We check the effect of information compression degree $\beta$ in $\mathcal{L}_{MI}$ with different environment difficulties in Table 2, and find that in simple tasks such as 1-lane TJ, the coach can handle $f^{-i}$ without obvious information compression. While in hard tasks, e.g., RC with $SR = 0.5$, a large $\beta$ can simplify communication messages, so that not to distract the agents and bring higher performance.

## 5.3 HARD TASK SMAC

We test CAMA's performance on the hard SMAC (Samvelyan et al., 2019) tasks with dynamic teams in this section. We use the setting from Liu et al. (2021) and Iqbal et al. (2021) that at the start of each episode a total of 3-8 agents are randomly divided into 2-4 groups and initialized at different places on the edge of a circle with the radius 9, and enemies are divided into 1-2 groups, Fig. 8(a). Agents have the sight range 9. Agents must learn to find teammates first before fighting against enemies with more quantities. We show the results of test win rate (TWR) on 3 maps in Fig. 8(b). Our method remarkably exceeds the current SOTA methods on all maps.

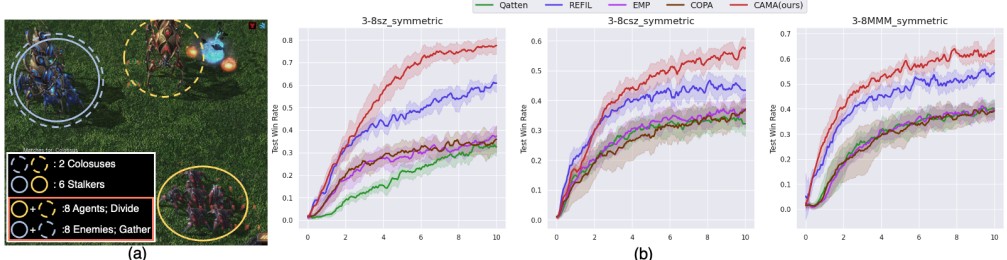

Figure 8: (a) An initialization teaser on SC2. (b) TWR comparisons on the 3 SC2 maps.

**Sight Range Testing.** In Fig. 9(a), with the varying SR comparisons in $\{3, 6, 9, \infty\}$, our CAMA reports superior TWR over SOTA methods on all SR settings and exhibits impressive robustness.

**Dynamic Team Composition.** In Fig. 9(b), we test the same model on different team sizes and plot the logarithm of the relative winning rate of the corresponding agent number against the total agents, where CAMA shows outstanding robustness to agent variation with a nearly stationary performance curve.

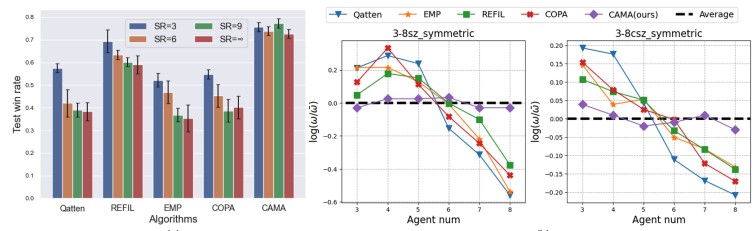

Figure 9: TWR comparisons with different SRs (a) and dynamic team composition (b). The x axis is the agent number which varies from 3-8. The y axis is $\log(\omega/\bar{\omega})$, where $\omega$ is the winning rate and $\bar{\omega}$ is agents' average winning rate. The black dotted line denotes $\bar{\omega}$.

## 6 CONCLUSION

In this paper, we explore the dilemma of the partial observability in MARL. With severe partial observability, agents usually can not sense the teammates and show poor team coordination. While with large sight range, agents are troubled by the attention distraction issue and exhibit apparent performance degradation. To tackle such sight range dilemma, we propose a Concentrated Attention for Multi-Agent reinforcement learning (CAMA) approach. First an Entity Dividing Module is used to divide raw entities for the local agents and the global coach separately. The suitable division percent varies due to the agents partial observability. For local agents, an Attention Enhancement Module improves their attention on execution-related entities under the condition of large sight ranges. For the global coach, the messages generated by novel conditional mutual information estimator replenish the information required for team coordination in all sight ranges. We evaluate our method on three commonly used MARL benchmarks: StarCraftII, MPE, and Traffic Junction. With raised agents' consciousness, the proposed CAMA reports significantly superior performance compared with SOTA methods and exhibits remarkable robustness to sight range variation and dynamic team composition.

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

## A  PROOF OF THE MASK GENERATOR

We now prove that $\mathcal{M}_s$ got by Eq. (5) is sufficient for Eq. (4).

We first show that $\mathcal{M}_f$ in Eq. (5) satisfies $||\mathcal{M}_f||_\infty \leq \alpha n_e$. Recall the definition of $\mathcal{M}_f$, it is a indicator matrix, that at each row, only the positions of top $\lfloor \alpha n_e \rfloor$ values of the corresponding row in the attention weight matrix are 1, and others are 0. Therefore, at each row of $\mathcal{M}_f$, there are at most $\lfloor \alpha n_e \rfloor$ 1s, which means the sum of the absolute value in each row is no larger than $\lfloor \alpha n_e \rfloor$. So $||\mathcal{M}_f||_\infty \leq \lfloor \alpha n_e \rfloor \leq \alpha n_e$. Then we show $||\mathcal{M}_s||_\infty \leq \alpha n_e$. Since the observability mask $\mathcal{M}$ is also a 0,1 mask, $||\mathcal{M}_s||_\infty = ||\mathcal{M} \odot \mathcal{M}_f||_\infty \leq ||\mathcal{M}_f||_\infty \leq \alpha n_e$.

Then we show that $\neg \mathcal{M}_s \odot \neg \mathcal{M} = \neg \mathcal{M}$, which equals to $\neg(\mathcal{M}_f \odot \mathcal{M}) \odot \neg \mathcal{M} = \neg \mathcal{M}$. Recall that $\mathcal{M}_f$ and $\mathcal{M}$ are both 0,1 matrices with the same shape. For any position in the matrices, we can use Table 4 to conclude all the situations:

| $\mathcal{M}$ | $\mathcal{M}_f$ | $\neg(\mathcal{M}_f \odot \mathcal{M}) \odot \neg \mathcal{M}$ | $\neg \mathcal{M}$ |
|---|---|---|---|
| 0 | 0 | 1 | 1 |
| 0 | 1 | 1 | 1 |
| 1 | 0 | 0 | 0 |
| 1 | 1 | 0 | 0 |

Table 4: Logical table of the mask.

We can find that at any situation $\neg(\mathcal{M}_f \odot \mathcal{M}) \odot \neg \mathcal{M} = \neg \mathcal{M}$, therefore $\neg \mathcal{M}_s \odot \neg \mathcal{M} = \neg \mathcal{M}$. In summary, $\mathcal{M}_s$ got by Eq. (5) is sufficient for Eq. (4).

## B  DERIVATION OF MUTUAL INFORMATION OBJECTIVE

We now give the derivation of Eq 7.

$$
\begin{aligned}
& \mathcal{I}(\zeta^i; f^{-i}) - \beta \mathcal{I}(\zeta^i; s) \\
=& H(\zeta^i) - H(\zeta^i|f^{-i}) - (\beta H(\zeta^i) - \beta H(\zeta^i|s)) \\
=& (1-\beta)H(\zeta^i) - (1-\beta)H(\zeta^i|s) + H(\zeta^i|s) - H(\zeta^i|f^{-i}) \\
=& (1-\beta)[H(\zeta^i) - H(\zeta^i|s)] - [H(\zeta^i|f^{-i}) - H(\zeta^i|f^i, f^{-i})] \quad \textcolor{magenta}{\text{//split } s \text{ into } f^i \text{ and } f^{-i}} \\
=& (1-\beta)\mathcal{I}(\zeta^i; s) - \mathcal{I}(\zeta^i; f^i|f^{-i}),
\end{aligned}
\tag{13}
$$

## C  PROOF OF THEOREM 4.1

For three random variables $\zeta^i$, $f^i$ and $f^{-i}$,

$$
\mathcal{I}_{CC}(\zeta^i, f^i|f^{-i}) \geq \mathcal{I}(\zeta^i, f^i|f^{-i}).
\tag{14}
$$

The equality holds if and only if $f^i$ is independent of the joint distribution of $\zeta^i, f^{-i}$.

*Proof.* Let $\Delta$ be the gap between $\mathcal{I}_{CC}(\zeta^i, f^i|f^{-i})$ and $\mathcal{I}(\zeta^i, f^i|f^{-i})$:

$$
\begin{aligned}
\Delta :=& \mathcal{I}_{CC}(\zeta^i, f^i|f^{-i}) - \mathcal{I}(\zeta^i, f^i|f^{-i}) \\
=& \left( \mathbb{E}_{\zeta^i, f^i, f^{-i}}[\log p(\zeta^i|f^i, f^{-i})] - \mathbb{E}_{f^i}\mathbb{E}_{\zeta^i, f^{-i}}[\log p(\zeta^i|f^i, f^{-i})] \right) \\
& - \left( \mathbb{E}_{\zeta^i, f^i, f^{-i}}[\log p(\zeta^i|f^i, f^{-i})] - \mathbb{E}_{\zeta^i, f^{-i}}[\log p(\zeta^i|f^{-i})] \right) \\
=& \mathbb{E}_{\zeta^i, f^{-i}}[\log p(\zeta^i|f^{-i})] - \mathbb{E}_{\zeta^i, f^{-i}}\mathbb{E}_f^i[\log p(\zeta^i|f^i, f^{-i})] \\
=& \mathbb{E}_{\zeta^i, f^{-i}}\left( \log[\mathbb{E}_{f^i}p(\zeta^i|f^i, f^{-i})] - \mathbb{E}_{f^i}[\log p(\zeta^i|f^i, f^{-i})] \right).
\end{aligned}
\tag{15}
$$

Since $\log(\cdot)$ is a concave function, $\Delta \geq 0$ due to Jensen's Inequality. $\square$

## D    GENERATE MESSAGES WITH MEMORY

In Sec. 4.4, we use $\mathrm{concat}(f^i, f^{-i})$ to generate the message $\zeta^i$. It means we have the true prior distribution $p(\zeta^i|f^i, f^{-i})$, which can be expediently applied to estimate $\mathcal{I}_{CC}$. However, if we wish to enhance the coach's ability, e.g., introducing a memory structure (similar to the GRU cell in local agents) into the message generator, the prior distribution turns into $p(\zeta^i|\boldsymbol{\rho})$ and therefore we can not obtain $p(\zeta^i|f^i, f^{-i})$ directly to estimate $\mathcal{I}_{CC}$. Similar to the idea in Cheng et al. (2020), we propose a variational term $q(\zeta^i|f^i, f^{-i})$ to estimate $p(\zeta^i|f^i, f^{-i})$:

$$\mathcal{I}_{vCC}(\zeta^i, f^i|f^{-i}) = \mathbb{E}_{\zeta^i, f^i, f^{-i}}[\log q(\zeta^i|f^i, f^{-i})] - \mathbb{E}_{f^i}\mathbb{E}_{\zeta^i, f^{-i}}[\log q(\zeta^i|f^i, f^{-i})]. \tag{16}$$

Theorem 1 gives a sufficient condition to ensure $\mathcal{I}_{vCC}(\zeta^i, f^i|f^{-i})$ be an upper bound of $\mathcal{I}(\zeta^i, f^i|f^{-i})$:

*Theorem* 1.  Denote $q(\zeta^i, f^i, f^{-i}) = q(\zeta^i|f^i, f^{-i})p(f^i)p(f^{-i})$. If

$$KL(p(f^i)p(\zeta^i, f^{-i})||q(\zeta^i, f^i, f^{-i})) \geq KL(p(\zeta^i, f^i, f^{-i})||q(\zeta^i, f^i, f^{-i})), \tag{17}$$

then $\mathcal{I}_{vCC}(\zeta^i, f^i|f^{-i}) \geq \mathcal{I}(\zeta^i, f^i|f^{-i})$. The equality holds when $f^i$ and the join distribution $\zeta^i, f^{-i}$ are independent.

*Proof.*  Let $\hat{\Delta}$ be the gap between $\mathcal{I}_{vCC}(\zeta^i, f^i|f^{-i})$ and $\mathcal{I}(\zeta^i, f^i|f^{-i})$. We have:

$$
\begin{aligned}
\hat{\Delta} =& \mathcal{I}_{vCC}(\zeta^i, f^i|f^{-i}) - \mathcal{I}(\zeta^i, f^i|f^{-i}) \\
=& \left( \mathbb{E}_{\zeta^i, f^i, f^{-i}}[\log q(\zeta^i|f^i, f^{-i})] - \mathbb{E}_{f^i}\mathbb{E}_{\zeta^i, f^{-i}}[\log q(\zeta^i|f^i, f^{-i})] \right) \\
& - \left( \mathbb{E}_{\zeta^i, f^i, f^{-i}}[\log p(\zeta^i|f^i, f^{-i})] - \mathbb{E}_{\zeta^i, f^{-i}}[\log p(\zeta^i|f^{-i})] \right) \\
=& \left( \mathbb{E}_{\zeta^i, f^{-i}}[\log p(\zeta^i|f^{-i})] - \mathbb{E}_{f^i}\mathbb{E}_{\zeta^i, f^{-i}}[\log q(\zeta^i|f^i, f^{-i})] \right) \\
& - \left( \mathbb{E}_{\zeta^i, f^i, f^{-i}}[\log p(\zeta^i|f^i, f^{-i})] - \mathbb{E}_{\zeta^i, f^i, f^{-i}}[\log q(\zeta^i|f^i, f^{-i})] \right) \\
=& \mathbb{E}_{f^i}\mathbb{E}_{\zeta^i, f^{-i}}[\log \frac{p(\zeta^i|f^{-i})}{q(\zeta^i|f^i, f^{-i})}] - \mathbb{E}_{\zeta^i, f^i, f^{-i}}[\log \frac{p(\zeta^i|f^i, f^{-i})}{q(\zeta^i|f^i, f^{-i})}] \\
=& \mathbb{E}_{f^i}\mathbb{E}_{\zeta^i, f^{-i}}[\log \frac{p(\zeta^i|f^{-i})p(f^i)p(f^{-i})}{q(\zeta^i|f^i, f^{-i})p(f^i)p(f^{-i})}] - \mathbb{E}_{\zeta^i, f^i, f^{-i}}[\log \frac{p(\zeta^i|f^i, f^{-i})p(f^i)p(f^{-i})}{q(\zeta^i|f^i, f^{-i})p(f^i)p(f^{-i})}] \\
=& \mathbb{E}_{f^i}\mathbb{E}_{\zeta^i, f^{-i}}[\frac{p(f^i)p(\zeta^i, f^{-i})}{q(\zeta^i, f^i, f^{-i})}] - \mathbb{E}_{\zeta^i, f^i, f^{-i}}[\frac{p(\zeta^i, f^i, f^{-i})}{q(\zeta^i, f^i, f^{-i})}] \\
=& KL(p(f^i)p(\zeta^i, f^{-i})||q(\zeta^i, f^i, f^{-i})) - KL(p(\zeta^i, f^i, f^{-i})||q(\zeta^i, f^i, f^{-i}))
\end{aligned}
\tag{18}
$$

$\square$

Theorem 1 reveals that $\mathcal{I}_{vCC}$ is an MI upper bound if the variational joint distribution $q(\zeta^i, f^i, f^{-i})$ is more "closer" to $p(\zeta^i, f^i, f^{-i})$ than to $p(f^i)p(\zeta^i, f^{-i})$. Let $q_\phi$ be the parameterization of $q$. In addition to the Eq. 16 that optimizes $\mathcal{I}_{vCC}$, we should minimize the KL divergence between $p(\zeta^i, f^i, f^{-i})$ and $q(\zeta^i, f^i, f^{-i})$:

$$
\begin{aligned}
& \min_\phi KL(p(\zeta^i, f^i, f^{-i})||q_\phi(\zeta^i, f^i, f^{-i})) \\
=& \min_\phi \mathbb{E}_{\zeta^i, f^i, f^{-i}}[\log \frac{p(\zeta^i|f^i, f^{-i})p(f^i)p(f^{-i})}{q_\phi(\zeta^i|f^i, f^{-i})p(f^i)p(f^{-i})}] \\
=& \min_\phi \mathbb{E}_{\zeta^i, f^i, f^{-i}}[\log p(\zeta^i|f^i, f^{-i}) - \log q_\phi(\zeta^i|f^i, f^{-i})] \\
=& \max_\phi \mathbb{E}_{\zeta^i, f^i, f^{-i}}[\log q_\phi(\zeta^i|f^i, f^{-i})],
\end{aligned}
\tag{19}
$$

Therefore, with sample pairs $\{(\zeta_k^i, f_k^i, f_k^{-i})\}_{k=1}^K$, we can maximize the log-likelihood of $q_\phi$:

$$\max_\phi \frac{1}{K} \sum_{k=1}^K \log q_\phi(\zeta_k^i | f_k^i, f_k^{-i}). \tag{20}$$

With enough optimization times of Eq. 20, $\mathcal{I}_{vCC}$ is guaranteed to be an MI upper bound.

We test the effect of coach with memory in the Resource Collection environment with $SR = 0.5$ and $SR = 1.0$, Table 5. We find that although equip the coach with the memory module improves the average performance, it brings large variance that the method becomes unstable. Therefore, to keep a stable performance of our method, we still use the coach with MLP in the main paper.

| Coach Style | Known Prior | SR=0.5 | SR=1.0 |
|---|---|---|---|
| MLP | $p(\zeta^i | f^i, f^{-i})$ | 385.25±25.61 | 458.13±46.93 |
| RNN | $p(\zeta^i | \boldsymbol{\rho})$ | 419.21±177.99 | 533.54±91.78 |

Table 5: Comparison on the style of coach.

# E  HEAT MAPS OF ATTENTION WEIGHTS

## E.1  AVERAGE HEAT MAPS OF MORE METHODS

As mentioned in Sec. 4.1, we visualize the attention weights of four methods in Fig. 10. Only our CAMA pay more attention to the agents themselves when the sight range is large, and therefore keeps a high performance.

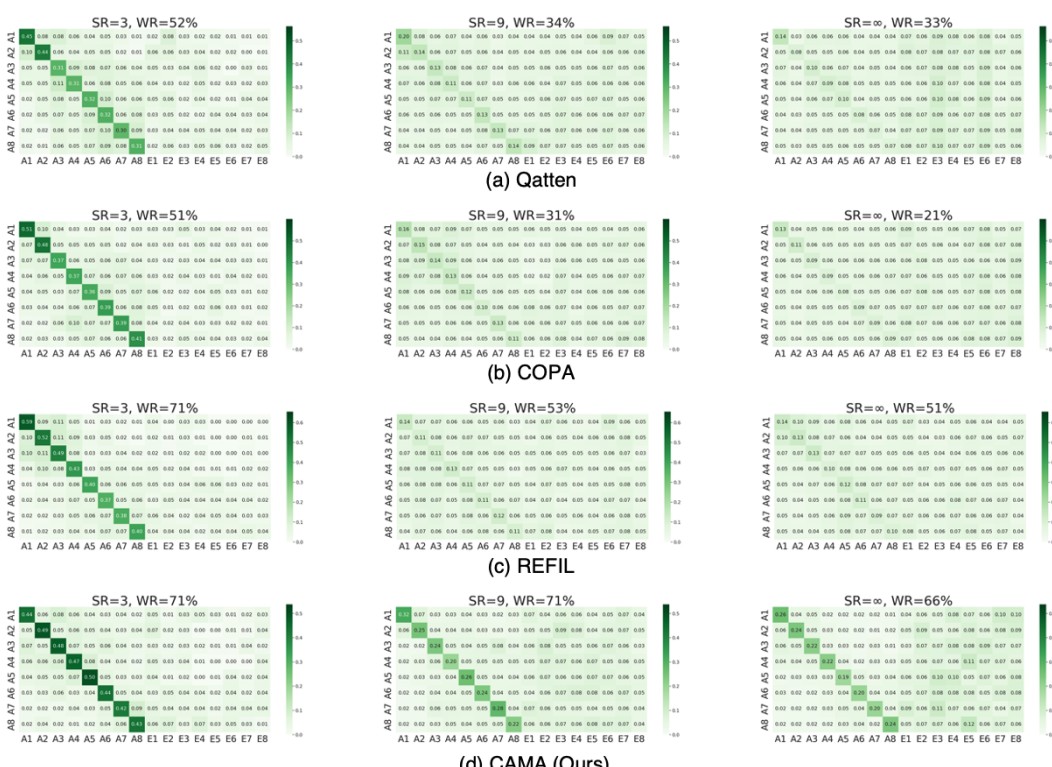

Figure 10: Attention Weights of Qatten, COPA, REFIL and our CAMA.

## E.2 Heat Maps at Single Time Steps

Figure 11: Attention Weights on single time steps of REFIL and our CAMA.

We also visualize the heat maps of our method and REFIL at single time steps. We choose the first attention head of each saved model, and set the color bar's range of attention weight to [0,0.35]. We visualize in Fig. 11 both methods' attention weights on all agents from $t = 0$ to $t = 24$, at which time all the agents are usually alive. We find that the large sight range distracts the attention of REFIL, while our method CAMA keeps attention concentration.

## F Hyperparameters

We summarize the hyperparameters in Table. 6

## G Environment Details

### G.1 Resource Collection

On a map of [-1,1], agents are initialized at random places with team size sampled from [3,8]. They need to collect resources from 6 resource points and transport the goods home. The locations of resource points and home are randomly sampled from the whole map at the start of each episode.

| Name | Description | Value |
|------|-------------|-------|
| $\gamma$ | Discounted factor | 0.99 |
| $\varepsilon$ anneal time | Time-steps for $\varepsilon$ to anneal from $\varepsilon_s$ to $\varepsilon_f$. $\varepsilon$ is the probability for agents choosing random actions. | 500000 |
| $\varepsilon_s$ | Start $\varepsilon$ | 1 |
| $\varepsilon_f$ | Final $\varepsilon$ | 0.05 |
| $n_{env}$ | The number of parallel environments | 8 |
| $|\mathcal{D}|$ | Replay buffer size | 5000 |
| $n_{head}$ | Number of heads in multi-head attention | 4 |
| $n_{attn}$ | Dimension of Attention embedding in local agent | 128 |
| $n_{ma}$ | Dimension of Attention embedding in mixer | 64 |
| $n_{rnn}$ | Dimension of RNN cells | 64 |
| $lr$ | Learning rate | 0.0005 |
| $\alpha$ | Default attention concentration rate | 0.8 |
| $\beta$ | Default communication message compression rate | 0.5 |
| $\alpha_{RMS}$ | $\alpha$ value in RMSprop | 0.99 |
| $\epsilon$ | $\epsilon$ value in RMSprop | 0.00001 |
| $n_{batch}$ | Batch size | 32 |
| $t_{target}$ | Time interval for updating the target network | 200 |
| $\lambda_1$ | Weight for $\mathcal{L}_{IM}$ | 0.005 |
| $\lambda_2$ | Weight for $\mathcal{L}_{MI}$ | 0.1 |
| $G_{max}$ | Clipping value for all gradients | 10 |

Table 6: Hyper-parameters.

The radius of the home and the resource location & agent is 0.1 and 0.05, respectively. There are 3 kinds of resources, and each agent $i$ has its own ability $b_i^e$ uniformly sampled from $\{0.1, 0.5, 0.9\}$ to collect each kind of resource $e$. Each agent can accelerate towards 4 directions or apply no forces at each time-step. Each agent has its maximal speed uniformly sampled from $\{0.3, 0.5, 0.7\}$ and the acceleration is fixed to 3.0. Every time the agent $i$ collects resource $e$, the team will get a reward $10 * b_i^e$. When an agent brings the resource home, the team will get a reward 1. An agent can only carry one resource at a time, which means an agent needs to bring the collected resource home before it starts to collect the next one. The episode limit is 145. The number of the agents for training is uniformly sampled from $\{2,3,4,5\}$, while for testing it is sampled from $\{6,7,8\}$. Each agent has a sight range $SR$. Entities including other agents and resource points that exceed agent $i$'s $SR$ are invisible to agent $i$.

## G.2 TRAFFIC JUNCTION

The simulated traffic junction environment from Sukhbaatar et al. (2016) has been a conventional and useful testbed for testing the performance of multi-agent communication algorithms Singh et al. (2018); Das et al. (2019); Liu et al. (2020). Despite its great success, cars in the original traffic junction environment can only move along pre-assigned routes on one or more road junctions and so the action space for each car only consists of two actions, i.e. gas and brake, which restricts its ability to simulate real-world environments and test communication algorithms. Moreover, the original observation is not fit for the need of entity-wise input.

So, we modify the original traffic junction environment to a more flexible version. In stead of pre-assigned routes, a random selected navigation target is assigned to each active car and the aim of each active car is navigating to its own target and avoiding collision which occurs when two cars are on same location. Along with the modified setting, the observation for each active car is altered to entity-wise form which contains the position of cars in a limited visibility (e.g. 3 x 3 for *sight_range* = 1) and the action space is more flexible with five actions i.e. *forward, back, left, right and wait*. Besides, the rewards consists of a linear time penalty $-0.01\tau$, where $\tau$ is the number of active time steps for the car since the last resurrection, a collision penalty $r_{collision} = -10$ and the difference Manhattan distance from the target between the previous and the current step . We choose two different difficulty levels following the settings in Singh et al. (2018), illustrated in Fig.12. Moreover,

| Map | $N_{max}$ | max_steps | $p_{arrive}$ start | $p_{arrive}$ end | entrances | targets |
|---|---|---|---|---|---|---|
| 1-lane | 5 | 20 | 0.1 | 0.3 | 4 | 4 |
| 2-lane | 10 | 40 | 0.05 | 0.2 | 8 | 8 |

Table 7: Detailed environment parameters in two versions of Traffic Junction Environment. $max\_steps$ is the length of each episode. $entrances$ or $targets$ indicate the number of choices of entrance or target and in the hard version the cars are restricted to keep to the right when entering or exiting, so there are only 8 entrances or targets.

the total number of cars is fixed at $N_{max}$ and the cars will be turned inactive when reaching the target or colliding with others and new cars get added to the environment with probability $p_{arrive}$ at every time-step which varies during training consistent with the curriculum learning in Singh et al. (2018). The detailed environment parameters settings are indicated in Table.7.

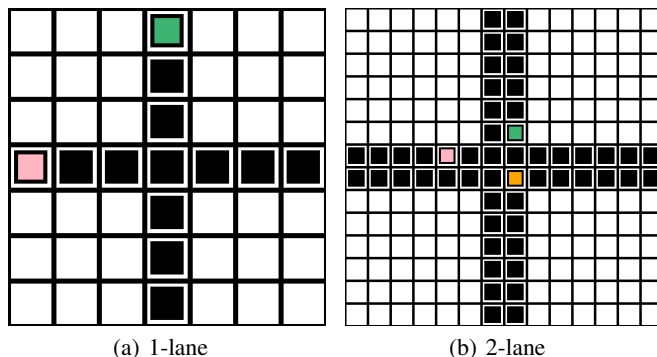

(a) 1-lane  (b) 2-lane

Figure 12: Traffic Junction Environment. Cars are navigating to their own assigned target and avoiding collision. There are two difficulty levels.

## H  HARDWARE

We ran experiments on 2 GPU servers, with each one having 8*RTX3090TI GPUS and 2*AMD EPYC 7H12 CPUs. Each experiment (one seed) takes 12-24 hours on one GPU.

## I  GENERALIZABILITY

We test the generalizability of our method in the environment Resource Collection. We train each method for $10^7$ time steps in the environment with 2-5 agents. Every $5 \times 10^4$ time steps, the model is evaluated on the environment with 6-8 agents for 160 episodes, and we plot the test return curves in Fig. 13. The results show that our method can generalize well to larger team sizes than training, which is probably due to the ARM module with the message generator that passes the global information.

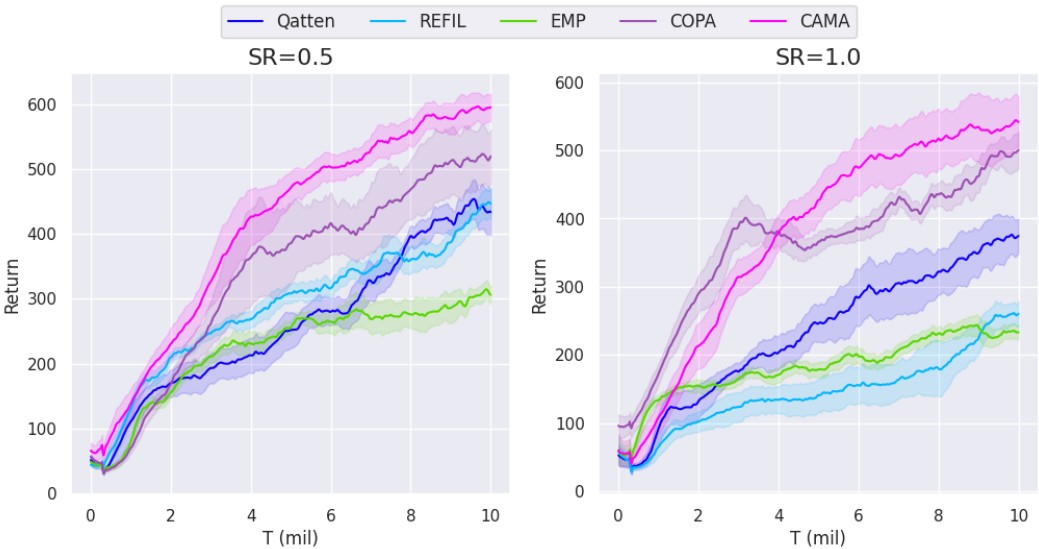

Figure 13: Generalizability on the task Resource Collection. Each method is trained on the agent number 2-5, and tested on agent number 6-8 every 50k time steps to plot the curve.

