# OpenReview forum: "Consciousness-Aware Multi-Agent Reinforcement Learning"
_ICLR.cc/2023/Conference — Submitted to ICLR 2023_

### Official Review · Reviewer_xWNF · 2022-10-20

**Confidence:** 3
**Correctness:** 1
**Technical Novelty And Significance:** 3
**Empirical Novelty And Significance:** 3
**Recommendation:** 1

**Clarity, Quality, Novelty And Reproducibility:**

Clarity: Some sections need more explanations and better organization.
Quality: The method section is good, the experimental section is average.
Novelty: It is an engineering enhanced version of multi-agent reinforcement learning. Maybe better than the threshold.
Reproducibility: They did not provide a link to the code.

**Details Of Ethics Concerns:**

The paper talks about "consciousness and awareness" when it does not really talk about this. This can be misleading making the people believe that computers are aware about itself. I consider it dangerous.

**Strength And Weaknesses:**

Strengths: Rigorous, style.
Weaknesses: Mainly the biggest is not knowing what is consciousness.

**Summary Of The Paper:**

Authors provide a method for "consciousness-aware" reinforcement learning that consists on several modules that provide attention and cooperation between agents to provide a solution for the varying partial observability. In particular, these modules are defined as "consciousness-aware" in an analogy to actions that a "consciousness-aware" human being do. However, we can not say that this system  possess awareness, nor phenomenal consciousness, just because of that. Hence, the title of the paper is misleading. It is a good paper but several terms must be changed. I would talk about attention or access consciousness but not awareness.

**Summary Of The Review:**

The main issue of this paper is their use of the words consciousness and awareness. Consciousness is a property that we cannot assign freely to machines without accepting lots of assumptions that can not be validated with the scientific method like multiple realizability and functionalism. Those can be reviewed in a philosophy of mind manual or even studied from a neuroscientific point of view. Even, Roger Penrose provides strong arguments against the emergence of consciousness in machines. Why is this algorithm in particular conscious, or aware, about itself? If it would, it need to catch qualia, and this assumption does not hold for a simple information processing algorithm such as the one present here. One thing is to talk about attention layers, which is valid, and another is just putting the title "Consciousness-aware multi-agent reinforcement learning". I can not accept the paper and need to recommend a strong reject, although technically is not that bad, just because of this. This seems to be a great multi-agent reinforcement learning algorithm from an engineering perspective but it is not aware of itself. Please review the literature about philosophy of mind to learn more about this. We can hardly assume that this algorithm would simulate features about access-consciousness, but not awareness which is the same as phenomenal consciousness. Consequently, the title can not include "consciousness" nor the name of the modules.

Outside of the discussion about consciousness:

Multiple terms such as varying partial observability vs slight partial observability need to be better defined.
I find a lack of cohesion between multiple parts of the paper, sections are written in a very straight-forward style. They need to be better introduced.
"Consciousness-learning" does not make sense.
The related work section deliberately ignores work on consciousness, which is OK if the paper does not talk about consciousness.
It would be interesting to describe an example of observation function.
Formalize the concept of dynamic team.
"mulit" -> multi
The experiment section is repeated just with 6 different seeds. I recommend to repeat it using 25 different seeds or more to make the results  statistically more solid.

---

> ### Author Response · Authors · 2022-11-08
> **Response to Reviewer xWNF**
>
> ### Q1: The term Consciousness-aware is not good.
> Thanks for the professional and constructive suggestion. We have prudently changed the term accordingly. Please refer to Common Issue Q1 for more details.
>
> ### Q2: Definition on varying partial observability/observation function.
> As suggested, we have added a clearer definition to Section 3, chapter "Multi-Head Attention (MHA) in MARL", and mark the content in blue. We consider a proximity-based observation function $Z(s^e, i)$ for agent $i$, i.e., each agent has a sight range $SR$, and $Z(s^e, i):=\\{ e^j|d(i,j)<=SR \\}, j\in[1,n_e]$, where $d(i,j)$ is the Euclidean (or Manhattan) distance between entity $i$ and $j$.
>
> ### Q3: The concept of dynamic team.
> The dynamic team composition means that at each episode, the team sizes may vary when agents join or quit the team at any time.
>
> ### Q4: More seeds in experiments.
> Since 5-8 seeds are commonly adopted in current MARL works, we choose 6 repeats under limited computational resources. To make the results more convincing, we will try our best to repeat the experiments for at least 2 more seeds.
>
> ### Q5: Cohesion between multiple parts.
> To emphasize the cohesion, we have changed some expressions in the abstract, introduction, method, and conclusion parts to clarify the connections between each module. We also summarize briefly here: First the Entity Dividing Module (EDM) split the raw entities into two parts according to each agent's observability. The first part is fed into the Attention Enhancement Module (ADM) to improve local agents' utility under large sight ranges by concentrating their attention on execution-related entities. The second part is fed into the Attention Replenishment Module (ARM), which is a coach with global sight, to summarize necessary information for agents' cooperation under low sight ranges. The ADM and ARM can be regarded as two parallel branches for processing information under different partial observability separately.

---

### Official Review · Reviewer_g4aW · 2022-10-22

**Confidence:** 4
**Correctness:** 3
**Technical Novelty And Significance:** 2
**Empirical Novelty And Significance:** 2
**Recommendation:** 5

**Clarity, Quality, Novelty And Reproducibility:**

Overall a good piece of work.  The presentation is clear; algorithms, modules, theories, experiments are all properly done.  It is just too bad that under the review framework, we cannot see live demos to evaluate its significances and pin down how and why the proposed CAMA is helpful.

The statement in the abstract is repeated also in the main text and conclusion.  It would be good to remark on the cost to pay and added complexity by adding CONSCIOUSNESS on top to coordinate all involved agents.

**Strength And Weaknesses:**

Strengths:
1. The problem is well stated, and it is a realistic problem to tackle.
2. The proposed method and its modules are well organized and proved.
3. The empirical study justifies the claim of SOTA against other methods.

Weaknesses:
1. The name CAMA uses the term consciousness but it is not clear what this term means in this context.  Consciousness is one of the most misunderstood terms. And when it is used, which parts of the system are unconscious?
2. Figures are well drawn and can be presented and explained in greater details.
3. It is not clear if CAMA requires higher volume of training data?

**Summary Of The Paper:**

This papers propose a “consciousness-aware” method to coordinate between multiple RL agents.  The authors state that individual may have attention distraction, and CAMA is good to address the issue to achieve better performance.

**Summary Of The Review:**

The paper is well structured and the proposed method and modules appear to be sound.  However, it is really hard to assess the impact of this work based on just a couple of examples.  It is not clear about its data complexity and limitations.  It is also good to thoroughly justify the use of the term “consciousness” and explain why “coordinated” or “supervised” can do a justified job.

---

> ### Author Response · Authors · 2022-11-08
> **Response to Reviewer g4aW**
>
> ### Q1: Consciousness problem.
> We remove the confusing term "Consciousness-aware" and use more specific "concentrated attention" instead. Please refer to Common Issue Q1 for details.
>
> ### Q2: Figures can be explained in greater details.
> Due to the limited space in the main paper, we add more specific descriptions of some experiments in Appendix G. If you are still confused about some figures, please point them out and we will try our best to give detailed explanations.
>
> ### Q3: Cost to pay and added complexity.
> Compared to traditional CTDE methods like Qatten, our method requires the same training samples (which can be seen from the x axis from the training curve of Fig. 5 and Fig. 8(b)). Since we add the communication module, our neural network have a bit more (+60%) parameters than baseline Qatten. And for the same experiment our method takes about 180% training times of the baseline. The extra cost on the communication module equips the agents with the ability to obtain out-of-sight-range information, and therefore can solve tasks that are hard for CTDE paradigm (Sec. 5.1, chapter "Coordination in Low Sight Ranges").

---

### Official Review · Reviewer_zvuV · 2022-10-22

**Confidence:** 4
**Correctness:** 2
**Technical Novelty And Significance:** 2
**Empirical Novelty And Significance:** 2
**Recommendation:** 3

**Clarity, Quality, Novelty And Reproducibility:**

The reviewer is aware that the authors describe a very complex work with many moving parts. However, it is very difficult to understand how the different mechanisms used in the paper are put together. The choice of the values of the different parameters is also not completely justified, except for some of them. In general, the paper appears hard to reproduce.

The reviewer has also concerns in relation to the originality of this work, since it appears as a composition of a series of different mechanisms that have been studied in the literature before. There is value in studying how to integrate different mechanisms, but in my opinion, the paper does not provide sufficient evidence for showing the contributions of each of them. The theoretical results presented in the work appear as somehow disjoint from the rest of the paper.

**Strength And Weaknesses:**

This paper considers the difficult problem of coordination of teams of agents for the execution of a complex task. The main mechanism used by the authors is the "classic" divide-and-conquer strategy. The goal is to deal with the problem of "attention distraction" in partially observed environments.

The use of term "consciousness-aware" is probably out of place, since the mapping to the concept of consciousness is actually very weak. Moreover, "consciousness" itself is about awareness, so the use of the term "consciousness-aware" might also be questionable. Some expressions should be really removed from the text such as "...CAMA empowers agents with purposeful consciousness". However, the reviewer believes that this is not the most important concern related to this work. One of the major weakness of this work is the fact that it looks like a composition of existing "components", which has somehow renamed and adapted. This would be fine, but the actual need for each of them is not supported by sufficient evidence.

The design of the different components is not completely new. They are essentially mechanisms that are mapped to existing work  in the area, which are correctly cited by the authors.

A key problem that is essentially not tackled in a general way is the decomposition of the general problem into subproblems. The presentation of these mechanisms is very high level in the paper. It is difficult to understand how this is performed in general (and I would say also in the environments considered by the authors).

The authors present some theoretical results about the "consciousness replenishment mechanism by local coach", but they appear rather disjoint from the main text. It is not completely clear why they are relevant for this work. In particular, the relevance of Theorem 4.1 to this specific work is not apparent. In fact, the theorem considers a property of a system of 3 random variables. How is this mapped to the situations and environments under consideration?

The reviewer would like to suggest the authors to improve their evaluation by showing the actual contributions of the different mechanisms separately (and in composition) from the point of view of the actual performance obtained in the system. Ideally, a discussion about the generalizability of the approach would really contribute to make this work stronger.



**Summary Of The Paper:**

The paper describes a solution for improving performance in multi-agent reinforcement learning. The term "consciousness" in this work essentially refers to attention based mechanisms and to the fact that the authors adopt and approach that resides in the idea of dividing the general problem into sub-parts in order to tackle it. The work is evaluated by means of a series of experiments with testbeds, namely an apple picking task with obstacles, a resource collection task, a traffic junction navigation task and the StarCraft environment.

The paper is evaluated using a variety of tasks and this is definitely commendable. However, the evaluation does not show the actual contributions of the three key components/mechanism of the proposed solution, namely the entity dividing module, the consciousness enhancement for the local agent and the consciousness replacement.

The main contribution of the work is essentially in putting together a series of existing mechanisms in order to achieve better performance in the tasks under consideration.




**Summary Of The Review:**

The contribution of this work is limited - the proposed solution is essentially a composition of existing components. This is not a problem per se, but the design choices and the actual evidence supporting the need for these components is not satisfactory. The evaluation is not satisfactory since it does not really show the contributions in terms of performance of the different mechanisms. The theoretical foundations of this work are also not completely convincing. The authors include some theoretical results that are not key for the actual contribution of this work. For this reason, the reviewer cannot recommend this work for publication.

---

> ### Author Response · Authors · 2022-11-08
> **Response to Reviewer zvuV**
>
> ### Q1: The term Consciousness-aware.
> As you suggested, we discard this dangerous term and turn to the "Concentrated attention" for clearer clarification. Please refer to Common Issue Q1 for details.
>
> ### Q2: The actual contributions of the three key components
> We clarify our ablation studies and the detail contributions of each component in Common Issues Q2. In summary, the Entity Dividing Module (EDM) structurally limit agent's observability when SR is large, and prepare entity embeddings for the latter two module. The  Attention Enhancement Module (AEM, previous CEM) focuses the agent's limited observability to execution-related entities and improve agent's utility on decentralized execution at any
> sight ranges. And the Attention Replenishment Module (ARM, previous CRM) introduces the communication mechanism to solve the out-of-sight-range cooperation problem when SR is small.
>
> ### Q3: The role of Theorem 4.1.
> Theorem 4.1 is used by ARM for generating communication messages. As we show in Sec. 5.1 "Coordination in Low Sight Ranges" chapter, without out-of-sight-range communication message, agents can not coordinate beyond their sight ranges. Theorem 4.1 is then developed to get a differentiable upper bound of our objective of message generation (Eq. (7) right side).
> And we show in Table 3 that messages generated by Theorem 4.1 can bring higher performance than other message generators. In addition, we give the reason why we do not use the left side of Eq. (7), which only requires the mutual information between two random variables. If we use the left side, we need to maintain a $p(\zeta^i|f^{-i})$ and a $p(\zeta^i|s)$ separately, which requires two MHA modules. An extra MHA module greatly increases the network parameters needed. In contrast, using the objective in right side accompanied with Theorem 4.1, we only need to maintain one distribution $p(\zeta^i|f^i, f^{-i})$, which is lightweight in network and can support larger training batch size compared to the left side.
>
> ### Q4: The main contribution of the work.
> Apart from arranging the existing tools in a novel fashion to solve the partial observability problem, we need to emphasize our following contributions: (1) We discover the "attention distraction" issue, and therefore uses the EDM to control agent's observability. Our operation on observability mask, as far as we know, is a novel way to deal with the multi-head attention module. (2) For ARM, we propose a new message generator by a new conditional mutual information estimator, rather than using the existing tools. Furthermore, we theoretically prove that our estimator upper bounds the true mutual information.
>
> ### Q5: A discussion about the generalizability of the approach.
> As suggested, we carry extra experiments on our methods' generalizability in Appendix I. The results show that our method can better generalize to larger team sizes than current SOTA.
>
> ### Q6: The method is hard to reproduce.
> As mentioned in Common Issues Q3, we have uploaded our code where one can reproduce our result and check implementation details.

---

> > ### Comment · Reviewer_zvuV · 2022-12-04
> > **Thanks for the clarifications**
> >
> > Many thanks for your clarifications - some notes about your replies:
> >
> > Q1. Avoid the terms consciousness is indeed a good choice.
> > Q2. It is still unclear if/how the three components interact in my opinion (do we need all of them? what are the contributions?)
> > Q3. That is an important clarification. However, I wonder if this is actually evaluated in the manuscript?
> > Q4. The reviewer still believes that the actual contribution of the work is unclear (the aspects related to "attention distraction" appears to be very related to the specific architecture).
> > Q5. There is limited evidence of the generalizability of the approach (also considering the experiments in the Appendix).
> > Q6. I would argue that the code is not sufficient in this case, since there are many aspects of the implementation that are very hard to understand from the text of the paper.
> >
> > I plan to keep my current score for this paper.

---

### Official Review · Reviewer_UtXQ · 2022-10-22

**Confidence:** 3
**Correctness:** 3
**Technical Novelty And Significance:** 2
**Empirical Novelty And Significance:** 3
**Recommendation:** 5

**Clarity, Quality, Novelty And Reproducibility:**

Clarity: The description of the problem setting and methods are relatively clear

Quality: It's not clear that the proposed architecture is solving the problems detailed by the introduction since the empirical justification is weak (see point 3 of weaknesses above) and a thorough ablation study is missing.

Originality: Most components of the method are similar to existing work, though they are arranged in a novel fashion to solve a novel problem.

Reproducibility: The architecture is complex and no code is provided, so this work may be difficult to reproduce.

**Strength And Weaknesses:**

# Strengths
* Strong results on a variety of domains. The consistent performance as the number of agents increases shown in figure 9bc is especially impressive. it would be interesting to see if this extends beyond the number of agents seen during training.

# Weaknesses
* The paper is rife with excessive anthropomorphizing language, down to the title ("consciousness"). I would recommend that the authors tone this down.
* The "consciousness replenishment module" is conceptually similar to the coach in COPA [1], though it is presented as a novel contribution with significant exposition dedicated to it. It should be described more in relation to the existing work to emphasize what the differences are.
* Figure 3 is averaging attention across time steps and (presumably) attention heads, so it would be expected to be dispersed over the whole set of agents and entities (since many varying interactions may happen over the course of an episode). If one were to plot attention at a single time step from a single head and it still looked this dispersed, it would be a more effective argument for what the authors are describing. The reason it looks less dispersed in the SR=3 case is likely that agents spend many time steps in this setting only being able to observe themselves.
* A thorough ablation study is needed to understand the significance of each component. For example, the loss functions that are unique to each module could be ablated.

# Questions
* From Figure 2, it seems like the entities in the complement of the selected mask are still used in computing the agent utilities, so why are they described as "out-of-sight-range"?

[1] Liu, Bo, et al. "Coach-player multi-agent reinforcement learning for dynamic team composition." International Conference on Machine Learning. PMLR, 2021.

**Summary Of The Paper:**

This paper presents an architecture which divides entities into two groups, one of locally relevant entities passed directly to the agent's utility network and another of less relevant entities which are passed into a global coach which compresses global information. This architecture produces improved performance on multi-agent tasks with varying quantities of agents.

**Summary Of The Review:**

While the results presented are impressive, the proposed architecture is complex and could use more justification for the individual components. It's not clear to me that the improved results are a result of addressing the problems laid out in the introduction or some other emergent properties of the architecture.

---

> ### Author Response · Authors · 2022-11-08
> **Response to Reviewer UtXQ**
>
> ### Q1: The term "Consciousness".
> We change the term as the reviewers suggested. Please refer to Common Issue Q1 for details.
>
> ### Q2: The "CRM" is conceptually similar to COPA.
> The only similar part of our method and COPA is that the structure uses the global state as input and outputs a vector for local agents. However, as we stated in related works "Communication Mechanism" chapter, this mechanism is widely used in many methods, and the optimization objective plays the vital role in the quality of the generated messages. Intuitively, our method aims to summarize the out-of-sight-range information at current state, for agents' information completion, while COPA wishes the message contain the information in future global trajectories. Methodologically, COPA needs to **maximize** some mutual information, therefore optimizing the lower bound, while part of our objective needs to **minimize** some conditional mutual information (Eq. (7) right side), and we develop a differentiable upper bound, which is completely opposite to COPA’s. In experiments, we summarize four methods using the centralized coach with different optimization objectives in Table 3, and our methods is far superior to the other three including COPA.
>
> ### Q3: Complementary entities are used for CRM, why are they described "out-of-sight-range"?
> Our optimization objective in Eq.(7) shows that we need to minimize the mutual information between the message $\zeta$ and the global state $s$. The complementary entities $f^{-i}$ is used for this. We use $f^{-i}$ as input to ensure that the output message contains the least information in $f^{-i}$.
>
> ### Q4: Per time step heat map.
> We visualize the heat maps of our method and REFIL at single time steps with the first attention head in Appendix E.2 as you suggested. The result is similar to the averaged heat maps and still support our assumption. In addition, we agree with your assumption on the less dispersed heat map on SR=3. Actually, that's the purpose of our experiment: to show that reducing the observability of the agent may help it learn better.
>
> ### Q5: Ablation study.
> Please refer to Common Issues Q2.

---

### Author Response · Authors · 2022-11-08
**Response to Common Issues.**

### Q1: The term "Consciousness-Aware".
Thanks for the kind reminder. Through repeated confirmation, we believe the misuse and overclaim of "consciousness" are mainly responsible for the degraded reading experience. We are sincerely sorry about the negligence that "attention" may be a more suitable substitution. We have revised the title to "Concentrated Attention for Multi-agent Reinforcement Learning" and polished our manuscript seriously to make it conceptually consistent and more readable (cf. blue texts in abstract, introduction, method, and conclusion).

For clarity, we briefly restate our core idea as follows: Agents' attention should be properly concentrated in specific partial observability. Under large sight ranges, an agent's attention should be limited to entities it can influence to avoid the ``attention distraction". Therefore, we constraint agents' observability by Entity Dividing Module (EDM) and flexibly forces their attention to execution-related entities by Attention Enhancement Module (AEM, previous CEM). Under low sight ranges, agents can not pay attention to unseen potential cooperators, so we use a coach with global sight to replenish the necessary information (which is distilled from the another branch of EDM's output) by Attention Replenishment Module (ARM, previous CRM). By dividing the entities and processing them in two parallel branches, our method shows stable performance and sustainable teamwork under complicated environments with varying partial observability. We hope this work can stimulate more interest on the study of partial observability in MARL.

### Q2: Lack of Ablation.
Actually, the whole Sec. 5.2 details our ablations on the separated contributions of the three parts of our method. To make it clearer, we have modified the linguistic expression. Besides, we briefly summarize the experiments and results here:

(1) Table 1 shows the contributions of AEM(previous CEM) and ARM(previous CRM) under different SRs, where $L_{IM}$ corresponds to AEM and $L_{MI}$ corresponds to ARM. Without $L_{IM}$, AEM degenerates into a common utility local Q network. Without $L_{MI}$, the communication message is replaced with a meaningless random vector, which equals to adding some noise dimensions to agents' local observations.

(2) Figure 6 shows the contributions of EDM. When $\alpha=1.0$, no extra observability constraint is made, which equals to replacing the EDM with the common observation function. The results show that a good $\alpha$ in EDM contributes a lot when SR is large.

(3) Apart from the contributions of the three modules, we also find the information compression degree $\beta$ in ARM plays an important role in different tasks(Table 2). And we show the superior performance of our new message generator in Table 3. We didn't show more results on the hyper-parameter of AEM because we find the loss is not sensitive to its weight, as long as its not too large to cover the RL loss. Maybe the reason is that $L_{IM}$ is a supervised loss and can be learnt far faster than $L_{QL}$.

---

### Decision · Program_Chairs · 2023-01-20

**Decision:**

Reject

**Justification For Why Not Higher Score:**

There was a clear lack of clarity with the original version of the paper. The original title of the paper probably also did not help.
I believe that the paper will benefit from a round of revisions and from more through empirical evaluations / ablations.

**Justification For Why Not Lower Score:**

NA

**Metareview: Summary, Strengths And Weaknesses:**

This paper suggests a novel attention mechanism to speed up and improve the final policy in multi-agent settings.
The main challenge for the paper is the lack of clarity and a lack of detailed ablations investigating the method. There are a lot of moving parts and it would be great to see which ones of these are actually required.